# Membership Inference Attacks Against Semantic Segmentation Models

## Abstract

Membership inference attacks aim to infer whether a data record has been used to train a target model by observing its predictions. In sensitive domains such as healthcare, this can constitute a severe privacy violation. In this work we attempt to address the existing knowledge gap by conducting an exhaustive study of membership inference attacks and defences in the domain of semantic image segmentation. Our findings indicate that for certain threat models, these learning settings can be considerably more vulnerable than the previously considered classification settings. We additionally investigate a threat model where a dishonest adversary can perform model poisoning to aid their inference and evaluate the effects that these adaptations have on the success of membership inference attacks. We quantitatively evaluate the attacks on a number of popular model architectures across a variety of semantic segmentation tasks, demonstrating that membership inference attacks in this domain can achieve a high success rate and defending against them may result in unfavourable privacy-utility trade-offs or increased computational costs.

## 1 Introduction

The ability of the ML community to train accurate, well-generalisable models can be significantly hindered by a lack of available data. One of the main reasons behind this lack of data is the perceived risk to data privacy, which reduces the willingness of individuals to contribute their data for scientific study. One example of adversarial involvement that can pose such a risk is a *membership inference attack* (MIA) (Shokri et al., 2017). This attack aims to identify if an individual data record was used to train the target model. The core idea behind MIA is the ability of an ML model to memorise the features associated with an individual data record. This results in the target model behaving differently (e.g. by having a higher confidence) when producing predictions on the data that it has previously been trained on, in comparison to the data it has not previously observed. This, in turn, can be leveraged by the adversary to infer if a specific record (that they obtain in advance) was used as part of the training dataset for a given ML task.

One feature of MIA which makes it particularly powerful is the flexibility of its threat model. Prior works in the area have explored a number of adversarial entry points ranging from an external black-box (BB) adversary with label-only model prediction access (Choquette-Choo et al., 2021) to an active white-box (WB) attacker that can influence the training procedure to their benefit (Tramèr et al., 2022). As a result, the research community faces a number of challenges when designing defences against MIA that can target a variety of threat models and not have a significant impact on the performance of the target model.

This, however, can prove to be a problematic task as –to-date– MIA has only been well-studied in the context of image classification (Shokri et al., 2017; Yaghini et al., 2022; Salem et al., 2018; Leino & Fredrikson, 2020; Yeom et al., 2017; Choquette-Choo et al., 2021). There exist a large number of works which exploit distinct characteristics of the training setting such as the loss value (Yeom et al., 2017), confidence intervals of the trained model (Shokri et al., 2017) and even the output labels of individual predictions (Choquette-Choo et al., 2021). However, so far, there has been little work in other ML domains that employ sensitive data and can be extremely vulnerable to these attacks, such as semantic segmentation. Given the increasing interest in segmentation models across many fields, (e.g. for medical diagnosis (Sheller et al., 2018; Chen et al., 2019;

Tajbakhsh et al., 2020; Li et al., 2020; Asgari Taghanaki et al., 2021; Valanarasu et al., 2021) or recognition of biological features (Hofbauer et al., 2019; Benini et al., 2019; Liu et al., 2020a; Yousaf et al., 2021; Li et al., 2021d)), many of which rely on sensitive patient data, we identify a misalignment of the existing adversarial capabilities and the existing research in this area. In this work, we attempt to address this knowledge gap by performing an in-depth investigation of MIA in the domain of semantic segmentation across a variety of threat models. In order to analyse the attackers of different relative strengths, we employ two main threat models: (i) A BB access adversary, who is able to query the target model, but cannot participate in its training and (ii) an active (or malicious) adversary, who can directly alter the data used to train the model to their advantage. For the latter task the adversary performs a backdoor attack (Chen et al., 2017): They craft a number of adversarial samples which are inserted into the training dataset.

We summarise our contributions as follows:

- We quantitatively evaluate membership inference attacks on popular models employed for semantic segmentation in binary and multi-class settings

- We perform an empirical comparison of the existing defences against membership inference attacks in these segmentation settings

- We investigate the effect of a more permissive threat model on the of MIAs under a malicious adversary employing poisoning attacks.

## 2 Related Work

### 2.1 Membership Inference Attacks

The first and one of the most widely employed implementations of membership inference attacks was proposed by Shokri et al. (2017). The main idea is to train a "shadow model" that replicates the behaviour of the target model. The adversary is then able to observe the outputs of the shadow model (distinguishing between the samples that were or were not included in its training set). The outputs, together with the target labels are combined with the known membership labels to produce a set of records, which is then used to train a binary classifier to determine the membership status of the victim data point(s). This type of attack typically requires the adversary to train a binary classifier that outputs a "member"-"non-member" prediction, but recent studies demonstrated that just observing the target model loss (Yeom et al., 2017), prediction confidence (Salem et al., 2018) or prediction entropy (Song & Mittal, 2021) can be enough to successfully infer membership. He et al. (2020) presented the first study of membership inference attacks on semantic segmentation models, which indicated that these models are also susceptible to such attacks. However, their experiments did not investigate MIAs on binary segmentation models, complex defences, or the effects of model poisoning. Additionally, we note that –while binary segmentation can be thought of as a subset of multi-class segmentation– attacks on these models can be more challenging to execute in practice compared to attacks on their multi-class counterparts. This is because a larger number of classes can leak more information about the underlying data (as each class corresponds to a segmentation mask, revealing much more semantic information), which is also the case for classification models (Shokri et al., 2017). The attack assumes training of a per-patch adversary that uses a structured loss map calculated between the model output and ground-truth label as its input. The evaluated defences include label-only outputs, simple regularisation techniques (such as an addition of Gaussian noise or dropout), and the use of differentially private stochastic gradient descent (DP-SGD) (Abadi et al., 2016), which adds Gaussian noise to the clipped gradients during training to protect privacy and prevent information leakage. According to the conducted experiments, label-only and regularisation defences could not sufficiently mitigate the attack risks and DP-SGD was the only method that could protect the membership status.

One of the underlying reasons behind the success of MIAs is overfitting (Shokri et al., 2017; Salem et al., 2018; Yeom et al., 2018; Chen et al., 2020; Leino & Fredrikson, 2020). This implies that the model "imprints" the information contained in individual training data samples into the model, producing predictions of higher confidence on the previously seen data samples. To address this challenge, a number of defences were proposed (particularly against black-box (BB) attacks, where an adversary has access only to the model outputs),

which either employ prediction masking (Shokri et al., 2017; Salem et al., 2018), differentially private training (Dwork, 2008; Abadi et al., 2016) or regularisation methods to mitigate the effects of overfitting (Li et al., 2021a; Hu et al., 2021; Hui et al., 2021; Kaya & Dumitras, 2021; Shejwalkar & Houmansadr, 2021). While defences such as differentially private training and regularisation are often effective, they usually do not provide satisfactory privacy-utility trade-offs. Adversarial regularisation was proposed by Nasr et al. (2018) to prevent membership inference attacks through an inclusion of an additional term in the loss function during target model training, which penalises the model for improving the success rate of the adversary. Therefore, the objective of such training is to simultaneously minimise the loss on the training data and the accuracy of the attack model.

Another method proposed by Shejwalkar & Houmansadr (2021) as a countermeasure against MIA is a specific instance of knowledge distillation (Hinton et al., 2015). The proposed approach is termed Distillation For Membership Privacy (DMP). DMP relies on two datasets, namely a labeled and an unlabeled (reference) one ($\{X_{tr}, Y_{tr}\}$ and $\{X_{ref}\}$ respectively). Once a large model is trained on the labeled data, this model is then used to predict the labels $\theta_{up}^{X_{ref}}$ of the unlabeled dataset. DMP then selects samples from the reference dataset with a low prediction entropy to train the new (protected) victim model. The intuition behind this defence is that using a reference dataset restricts access to the private labeled training data, discouraging the memorisation of features associated with individual data points, reducing the membership information leakage.

## 2.2 Backdoor Attacks

Another class of attacks, namely model poisoning attacks concentrate on altering the functionality of the trained model. Typically, these are executed by the attackers who can influence the training procedure and embed a number of malicious data samples into the training dataset (e.g. by introducing an incorrect label (Paudice et al., 2018)). These attacks can result in a deliberate utility reduction for a specific class of data (Shafahi et al., 2018), evasion of detection at inference time (Huang et al., 2020) or even in an introduction of imperceptible background (and adversarially-controlled) auxiliary tasks (Chen et al., 2017). The last interpretation of such attack is typically termed a "backdoor attack".

The aim of backdoor attacks on deep neural networks is to inject a "trigger" (i.e. a pixel pattern) into selected training data samples to manipulate the behaviour of the attack model on testing data with such trigger (Gu et al., 2019). Shokri et al. (2020) and Lin et al. (2020) investigate the extensions of such attacks, where the signals of the malicious inputs are hidden in the latent representation and the poisoned samples include a specific combination of visual features, which renders backdoor detection algorithms ineffective. To conceal the attack better and to prevent the recognition of the poisoned samples, recent works explore invisible triggers based on noise addition (Li et al., 2021c; Doan et al., 2021) or image warping (Nguyen & Tran, 2021), highlighting the potential dangers associated with this class of adversarial influence.

Similarly to prior works on MIA, the existing backdoor attacks have been studied primarily in classification settings (Gu et al., 2019; Yao et al., 2019; Liu et al., 2020b; Li et al., 2021c; Nguyen & Tran, 2021) where the objective is to predict a single label given the input. However, recently empirical comparisons of backdoor attacks in the domain of semantic segmentation (Li et al., 2021b; Feng et al., 2022) have emerged. Overall, the investigation of these attacks in the domain of semantic segmentation has been limited, as the prior works lack an evaluation of the interdependency between the poisoning rate and the success of the attack, as well as them relying on large, contrastive non-semantic triggers which may limit the applicability of these works.

## 3 Methodology

### 3.1 Types of Membership Inference Attacks

To evaluate the threat of MIAs on segmentation models we investigate the following attacks: (i) The **Type-I attack** uses $n$ shadow models and a single attack model, which itself is a binary classifier that uses only the output of the victim model to distinguish between member and non-member data samples. The intuition

behind the attack is that the segmentation masks will have more refined features if the input came from a private training dataset and it has been previously seen by the victim model. The aim of the attack model is to learn those differences and disclose the membership information; (ii) The **Type-II attack** is inspired by the membership inference attack proposed by He et al. (2020) as it uses the victim model outputs as well as ground-truth masks that concatenated channel-wise into a single tensor. The inclusion of ground-truth masks gives the attack model more information about the accuracy of the prediction to improve the membership inference performance. The attack is otherwise the same as the Type-I attack; (iii) The **Global loss-based attack** aims to reveal the membership information only based on the loss of the victim model given a single testing data sample. Unlike the previous attacks, it does not require training of the attack model, reducing the overall computational complexity. Inspired by attack introduced by Yeom et al. (2018), our global loss-based attack relies on a threshold determined as the mean of the shadow model training losses. If the loss of the victim on a data record is less than or equal to the threshold, it is classified as a member.

## 3.2 Defences Against Membership Inference Attacks

We empirically assess the effects of various defences against MIAs in semantic segmentation settings. It is of note that most of these methods have originally been proposed for classification models. The proposed defences are the following: (i) The **argmax defence** inspired by the label-only defence aims to reduce the amount of information included in the victim model outputs by returning a mask that contains only the predicted object labels instead of prediction distributions. For binary segmentation tasks, the pixel-wise probabilities are rounded to the nearest integer, and for multi-class settings, the argmax defence transforms model output to a single matrix where each element represents the predicted label; (ii) **Crop training** is a form of regularisation that mitigates the effects of overfitting. Once an image is sampled from the training dataset, the algorithm takes a crop from a random location in the image-mask pair. The defence does not allow the victim model to observe the whole input image, limiting the amount of information that can be imprinted into the model; (iii) The **mix-up defence** conceals the information in the training dataset by combining multiple image-mask pairs into a single one. Once a training batch $B$ is sampled, it is then copied $n-1$ times where each copy $B^{(m)}$ is randomly permuted, resulting in $n$ distinct versions of the same batch. Next, an image-mask pair $B_i^{(m)}$ is taken from each batch. Those $n$ pairs are combined into a single pair by taking a set of random vertical slices from images $[B_i^{(m)}]^x$ and their corresponding masks $[B_i^{(m)}]^y$, and appending them together to create a new image-mask pair $\{x', y'\}$ of the same dimensions as the images from the original dataset. Algorithm 1 describes the method for $n = 2$ where $[A]_{:i}$ denotes the $i$th column of a matrix $A$; (iv) The **min-max defence** employs adversarial regularisation with the objective to reduce the distance between the outputs of the member and non-member samples. During training of the victim model, an additional classifier that acts as an adversary is trained in parallel. The accuracy of this classifier is then used as regularisation term for the victim model. We assume that the classifier employs a Type-II attack to distinguish between the member and non-member data records; (v) **Knowledge distillation** defence replicates the method introduced by Shejwalkar & Houmansadr (2021) for classification models. However, the assumption on the teacher model outputs is more relaxed as the predicted masks are modified to only include the predicted labels. While the prediction probabilities are removed, the structures of segmentation masks are kept intact. The data samples used to train the protected model are those for which the teacher model returns an output with a loss smaller than or equal to its validation loss.

## 3.3 Backdoor Attacks

We evaluate the success of training data poisoning based on two trigger shapes, a line and a square, as demonstrated in Figure 1. The line trigger is placed on the top of the poisoned image spanning the entire width, with a height of one pixel. The square trigger is $3 \times 3$ pixels large and is always placed in the top left corner of the poisoned image.

**Input:** Batch of $k$ image-mask pairs $B = [\{x_1, y_1\}, ..., \{x_k, y_k\}]$, image dimensions $H \times W$.
Initialise an empty array $B'$
**Create permutations**
$B^{(1)} = permute(B)$
$B^{(2)} = permute(B)$
**Get a splitting point**
$b \sim \text{Beta}(\alpha = 2, \beta = 2)$
$\gamma = \lceil W \cdot b \rceil$
**Mixing-up permuted batches**
**for** $i \in \{1, ..., k\}$ **do**

$\quad x' = \left[ [B_i^{(1)}]_{:1}^x, ..., [B_i^{(1)}]_{:(\gamma-1)}^x, [B_i^{(2)}]_{:\gamma}^x, ..., [B_i^{(2)}]_{:W}^x \right]$

$\quad y' = \left[ [B_i^{(1)}]_{:1}^y, ..., [B_i^{(1)}]_{:(\gamma-1)}^y, [B_i^{(2)}]_{:\gamma}^y, ..., [B_i^{(2)}]_{:W}^y \right]$

$\quad$ Append mixed-up image-mask pair $\{x', y'\}$ to $B'$
**end**
**Output:** Batch of mixed-up image-mask pairs $B'$.

**Algorithm 1:** Mix-up batch pre-processing for $n = 2$. Beta indicates the beta distribution with parameters $\alpha$ and $\beta$.

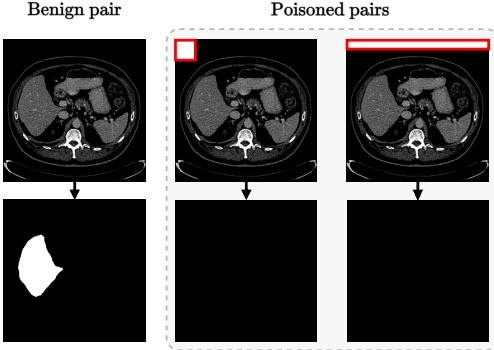

Figure 1: Demonstration of benign and poisoned image-mask pairs from the MSDD (Simpson et al., 2019) with exaggerated square and line triggers highlighted with a red line.

## 4 Threat Model

For a general MIA setting, it is assumed that an adversary is honest-but-curious (HbC) in a BB setting, where the only information about the victim model can be accessed through its outputs. We assume the adversary has access to a shadow dataset that comes from the same distribution as the private dataset used by the victim (the intersection of both datasets is an empty set). The adversary then trains a single shadow model to mimic the behaviour of the victim model. We consider the adversary has an access to the details of the victim model hyperparameters which are then applied during training of the shadow model.

In contrast, for our evaluation of MIA together with backdoor attacks, we assume that the attacker is malicious and has access to the data loader used to train the model. The poisoned images are stamped with a trigger and the objects on the corresponding segmentation masks are removed before being passed to the victim model.

## 5 Experimental Setup

In this section we introduce the datasets and the experimental setups for our evaluation of membership inference attacks in binary and multi-class segmentation settings.

## 5.1 Datasets

We use three datasets to investigate the feasibility of membership inference attacks on segmentation models: (i) The Medical Segmentation Decathlon Dataset (MSDD) (Simpson et al., 2019), which consists of manually annotated medical images of various anatomical sites for benchmarking of segmentation methods. It includes $2,633$ three-dimensional computed tomography (CT) and magnetic resonance imaging (MRI) scans collected from multiple sources representative of the real-world clinical applications; (ii) The Kvasir-SEG Dataset (Pogorelov et al., 2017) is an open-access dataset of gastrointestinal polyp images and corresponding segmentation masks annotated and verified by an experienced gastroenterologist. The dataset itself contains $1,000$ polyp images and their corresponding ground-truth segmentation masks with varying resolutions; (iii) The Cityscapes Dataset (Cordts et al., 2016) contains a set of images and corresponding segmentation masks for tasks related to urban scene understanding. It includes $5,000$ images taken on streets of 50 different cities with high-quality pixel-level annotations, and $20,000$ additional coarsely annotated images.

## 5.2 Membership Inference Attacks on Binary Segmentation Models

This subsection describes the experimental setup used to investigate the performance of membership inference attacks and the defences against them in a binary segmentation domain.

### 5.2.1 Model Dependency

The success of membership inference attacks was evaluated in model-dependent and model-agnostic settings. In the former setting, the shadow model architecture was identical to the architecture of the victim model, whereas the latter setting allows a mismatch between the victim and the shadow architectures. Regardless of the setting, shadow models were trained with the same hyperparameters and training set sizes as the corresponding victim models.

### 5.2.2 Datasets and Architectures

Binary segmentation models were trained on a subset of liver images from MSDD (referenced below as the LIVER dataset) and the Kvasir-SEG Dataset. We employed a U-Net architecture with ResNet-34, MobileNetV2 and VGG11 as encoders for our segmentation models. When evaluating the knowledge distillation defence, the architecture of the protected model was the same as the architecture of the unprotected model. Every attack model was a ResNet-34 binary classifier initialised with weights using a pre-trained ImageNet-1K (Deng et al., 2009) classifier.

### 5.2.3 Setup for Data

Given a set of data samples, four different subsets were used to train a victim model $V$ and a shadow model $S$: $D_{train}^{victim}$, $D_{test}^{victim}$, $D_{train}^{shadow}$ and $D_{test}^{shadow}$. The shadow datasets were then merged into a single dataset that was inputted into the shadow model along with the corresponding membership labels to train the attack model $A$. We also used a fifth disjoint set $X_{ref}$ for experiments that evaluated the effects of knowledge distillation. We trained the segmentation models using the following configuration: (i) We used the LIVER dataset to obtain training and testing sets that consisted of 500 image-mask pairs and $X_{ref}$ with $1,000$ pairs. An attack model $A$ was trained on a set $D_{train}^{shadow} \cup D_{test}^{shadow}$ that included true membership labels, and then evaluated on $D_{train}^{victim} \cup D_{test}^{victim}$; (ii) Next, we evaluated MIA on models trained using the Kvair-SEG Dataset, where the size of the training and test datasets was set to 300 and 200, respectively. Moreover, when training and evaluating the attack model, a subset of 200 pairs was taken from the training sets to create balanced datasets of 400 pairs.

Each experiment was conducted independently and the data samples were randomly selected and distributed over the sets. We note that although it is customary to split patient data in clinical settings, for MIA we assume that the adversary has access to matching patient records for running the attack (in the case of experiments that employ the LIVER dataset). Each image and mask was resized to $256 \times 256$ pixels. When evaluating the crop training defence, we took crops of size $128 \times 128$ pixels from the resized image-mask pairs.

### 5.2.4 Setup for Training

The general segmentation model training setup (unless specified otherwise) can be found below. The number of epochs was set to 70, apart from the models with crop training-based defence which were trained for 210 epochs to mitigate the loss of information in batches that arises from cropping. Each batch consisted of 8 image-mask pairs and the learning rate was set to $1e{-}4$ aside from the models trained with DP-SGD for which the learning rate was increased to $4e{-}4$. For models trained in a deferentially private setting, target $\varepsilon$ and $\delta$ were set to 8.5 and $2e{-}3$, respectively. Models that employed the mix-up defence combined two image-mask pairs into one during training.

Every attack model was trained for 30 epochs with a learning rate $1e{-}4$ and a batch size of 4. The adversarial regularisation adopted the same learning rate and its weight coefficient in the loss function of the victim model were set to $\lambda = 0.005$ and $\lambda = 0.05$ for the LIVER and Kvasir-SEG training datasets respectively.

### 5.3 Membership Inference Attacks on Multi-Class Segmentation Models

The evaluation of membership inference attacks in a multi-class segmentation domain follows the same setting as described in Subsection 5.2 with the following exceptions: (i) Every segmentation model used the same U-Net architecture with a ResNet-34 encoder; (ii) The evaluation was performed on the Cityscapes dataset using only the high-quality image-mask pairs with pixel-level annotations. Each training and testing dataset consisted of 400 image-mask pairs and an attack model was always trained and tested on a balanced set of 800 image-mask pairs; (iii) All images and masks were resized to $512 \times 256$ pixels; (iv) Type-I and Type-II attack models were trained for 10 epochs.

### 5.4 Membership Inference Attacks and Backdoor Attacks

To investigate the effects of a malicious adversary with backdoor insertion capabilities on the success of MIA, we followed the same evaluation setting as described in Subsection 5.2. For the evaluation we used the LIVER dataset. We assumed a model-dependent setting in which every segmentation model used a U-Net architecture with a ResNet-34 encoder. The threat model allows this adversary to access the victim model data loader, where each training data sample can be poisoned with a certain probability. Whenever an image-mask pair was selected for poisoning, a trigger was applied to the image and the target object was removed from the corresponding mask.

The trigger values were set to 255 in order to maximise the contrast between the trigger and the image background, unless stated otherwise. However, we also evaluated the effects of altering the colour intensity of the triggers. We assumed two simultaneous attacks on the victim model, a backdoor attack to embed some hidden functionality during training and a subsequent membership inference attack on the deployed model. The success of MIAs was always measured on benign data only.

## 6 Experimental Results

### 6.1 Membership Inference Attacks on Binary Segmentation Models

The performances of Type-I, Type-II and global loss-based membership inference attacks (for segmentation model architectures trained on the LIVER dataset) are shown in Tables 1, 2 and 3, respectively (with a visual comparison in Figure 2). The Type-I MIA is the least successful with the highest achieved accuracy of 62.3%. For Type-II and global loss-based attacks, the top testing accuracy was 85.7% and 80.6%, respectively. Overall, Type-II and global-loss based attacks drastically outperform Type-I attacks (even in the worst case their accuracy was approximately 2% lower than the best performing Type-I attack). Regardless of the attack type, the highest accuracy was always achieved when the architectures of victim and shadow models matched.

Table 1: Comparison of Type-I attack performances (in %) on undefended binary segmentation models trained on the LIVER subset of MSDD in model-dependent and model-agnostic settings. Highlighted values indicate the best and the worst MIA performances.

| Victim model | Shadow model | Accuracy | F1-score |
|---|---|---|---|
| ResNet-34 | ResNet-34 | 60.9 | 51.2 |
| | MobileNetV2 | 60.6 | 70.0 |
| | VGG11 | 56.0 | 69.3 |
| MobileNetV2 | ResNet-34 | 55.2 | 26.6 |
| | MobileNetV2 | 62.3 | 65.5 |
| | VGG11 | 60.8 | 70.2 |
| VGG11 | ResNet-34 | 54.1 | 30.4 |
| | MobileNetV2 | 54.2 | 45.4 |
| | VGG11 | 61.3 | 49.2 |

Table 2: Comparison of Type-II attack performances (in %) on undefended binary segmentation models trained on the LIVER subset of MSDD in model-dependent and model-agnostic settings. Highlighted values indicate the best and the worst MIA performances.

| Victim model | Shadow model | Accuracy | F1-score |
|---|---|---|---|
| ResNet-34 | ResNet-34 | 85.7 | 83.7 |
| | MobileNetV2 | 69.0 | 72.8 |
| | VGG11 | 66.8 | 64.0 |
| MobileNetV2 | ResNet-34 | 64.4 | 73.3 |
| | MobileNetV2 | 65.6 | 70.5 |
| | VGG11 | 62.1 | 69.8 |
| VGG11 | ResNet-34 | 61.1 | 65.9 |
| | MobileNetV2 | 76.6 | 79.6 |
| | VGG11 | 85.3 | 84.4 |

Table 3: Comparison of global loss-based attack performances (in %) on undefended binary segmentation models trained on the LIVER subset of MSDD in model-dependent and model-agnostic settings. Highlighted values indicate the best and the worst MIA performances.

| Victim model | Shadow model | Accuracy | F1-score |
|---|---|---|---|
| ResNet-34 | ResNet-34 | 80.6 | 76.2 |
| | MobileNetV2 | 76.0 | 69.0 |
| | VGG11 | 68.1 | 54.2 |
| MobileNetV2 | ResNet-34 | 60.4 | 34.7 |
| | MobileNetV2 | 69.4 | 58.0 |
| | VGG11 | 62.4 | 40.1 |
| VGG11 | ResNet-34 | 73.4 | 64.0 |
| | MobileNetV2 | 75.4 | 67.7 |
| | VGG11 | 76.8 | 70.2 |

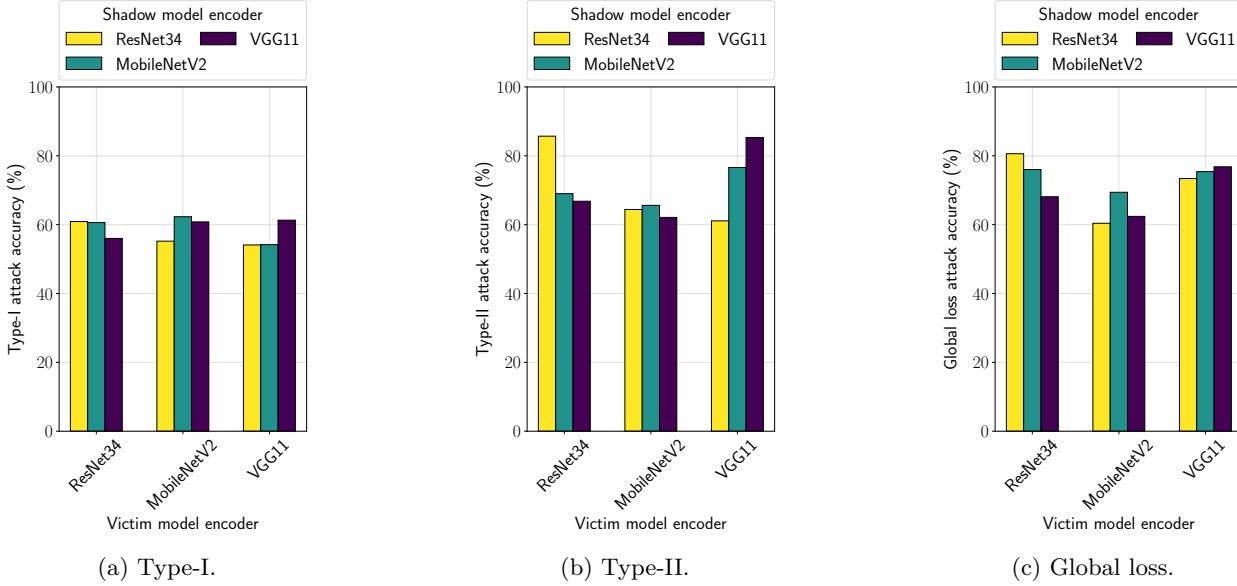

(a) Type-I.  (b) Type-II.  (c) Global loss.

Figure 2: Visual comparison of performances given various attack types and victim/shadow model combinations (based on Tables 1, 2 and 3).

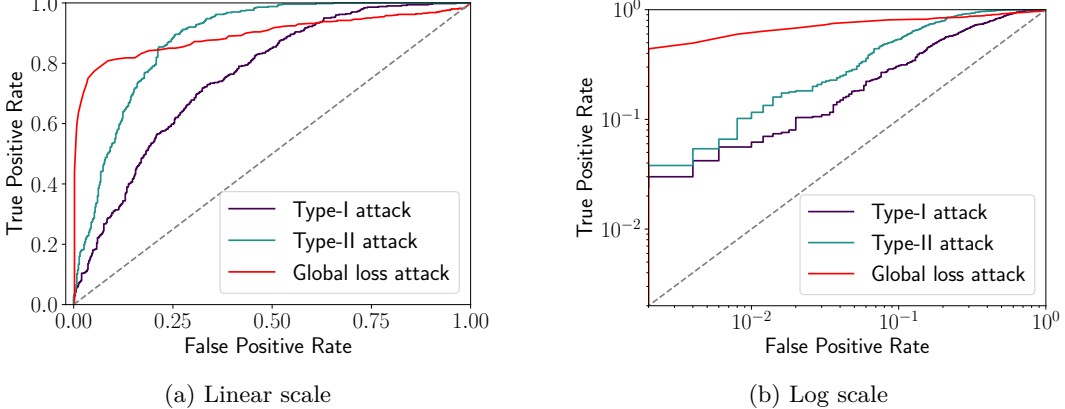

(a) Linear scale  (b) Log scale

Figure 3: The ROC curves for Type-I, Type-II and global loss-based membership inference attacks on undefended segmentation models shown with a linear scaling (left) and a log-log scaling (right) to emphasise the performance in the low-FPR range.

### 6.1.1 Low False Positive Rate Performance

The ROC curves of the Type-I, Type-II and global loss-based membership inference attacks on undefended segmentation models can be found in Figure 3. The global loss-based attack resulted in best performance at low false-positive rates ($\leq 0.1$), followed by the Type-II and Type-I attacks. However, for FPR $\geq 0.22$, Type-II attack achieved a higher true-positive rate compared against the global loss-based attack.

### 6.1.2 The Effects of Overfitting

In this Section, we investigate the effects of model overfitting on the success of MIAs. We explored two distinct settings: (i) A global loss-based attack on a set of victim models with a number of training epochs varying from 40 to 100 (with the shadow models being consistently trained over 70 epochs); and (ii) A Type-II attack where the training set size ranged from 500 data samples to 900 for both the victim and

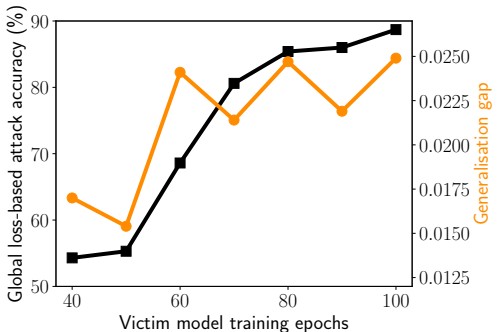

(a) Overview of the relationship between global loss-based attack accuracy and the size of a generalisation gap.

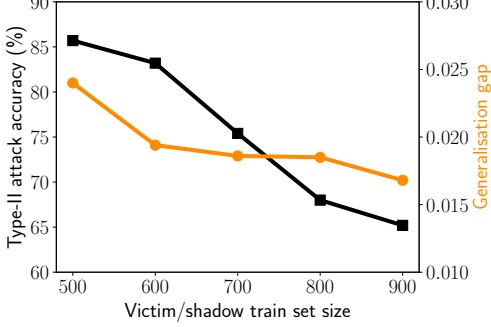

(b) Overview of the relationship between Type-II attack accuracy and the size of a generalisation gap (based on Table 4).

Figure 4: MIA accuracy given the size of a generalisation gap of a victim model measured as the difference between the mean Dice coefficients given training and testing sets.

Table 4: Overview of the relationship between Type-II attack performance (in %) and the generalisation gap of the victim model measured as the difference between the mean Dice coefficients of the training and the testing set. With an increase in the train set size, the effect of overfitting is not as pronounced because the final model generalises better, which consequently decreases the attack accuracy. No defence is used by victim models trained on LIVER dataset.

| Train set size | Generalis. gap | Accuracy | F1-score |
|---|---|---|---|
| 500 | 0.0240 | 85.7 | 83.7 |
| 600 | 0.0194 | 83.2 | 85.5 |
| 700 | 0.0186 | 75.4 | 70.7 |
| 800 | 0.0185 | 68.0 | 64.4 |
| 900 | 0.0168 | 65.2 | 62.6 |

shadow models. We measured the generalisation gap of the victim model as the difference between the mean Dice coefficients for the given training and testing datasets. Figure 6 shows the results of Type-II and global loss-based membership inference attacks.

The accuracy of the global loss-based attack increased from 54.3% to 88.7% when increasing the number of epochs the victim model was trained for (with the generalisation gaps of 0.017 and 0.0249 respectively). Additionally, increasing the size of the training set from 500 to 900 samples reduced the Type-II attack accuracy from 85.7% to 65.2%. We show the detailed performance evaluation for each training dataset size in Table 4.

### 6.1.3 Defence Methods

We summarise the impact of defences against Type-II and global loss-based attacks on segmentation models trained on the LIVER dataset in Table 5 and Figure 6. We additionally highlight the drops in utility caused by the applied defences.

The argmax defence decreased the attack accuracy by 18.3% and 5.3% for Type-II and global loss-based attacks, respectively, compared to attacks on unprotected models. Moreover, while it provided the model owner with protection against the Type-II attack, it offered the lowest level of protection out of all evaluated defences.

Crop training regularisation significantly reduced the accuracy of both Type-II and global loss-based attacks to 55.3% and 50.3% which, however, came with a cost of utility degradation for the victim model (as indicated

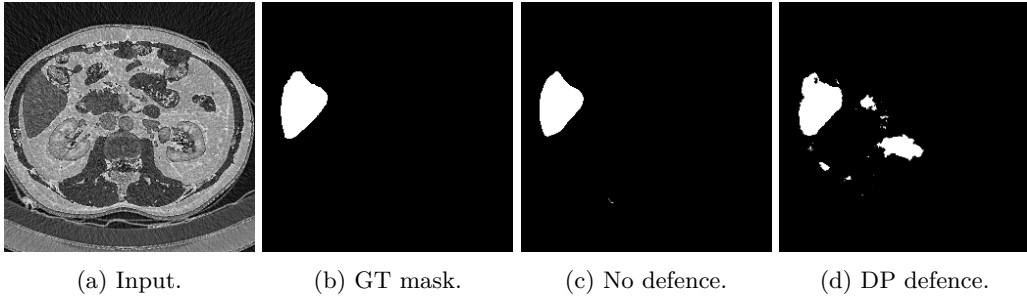

(a) Input.  (b) GT mask.  (c) No defence.  (d) DP defence.

Figure 5: Comparison of liver segmentation masks produced by an undefended model and a model trained using DP-SGD with the ground-truth mask (based on data samples from the MSDD dataset).

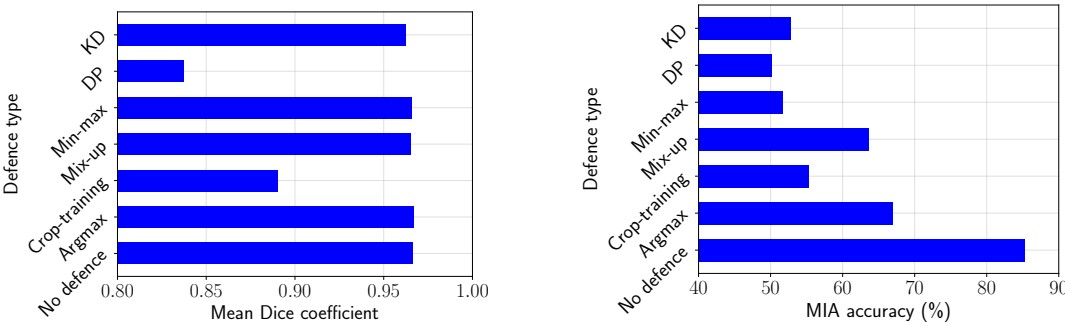

Figure 6: An overview of target model utility (top), measured as the mean Dice coefficient on a testing set, and the Type-II MIA success when employing various defence mechanisms during training (bottom).

in Figure 6). Mix-up defence reduced MIA accuracy by over 20% regardless of the attack formulation, while still offering a lower level of protection when compared against training with cropped image-mask pairs. However, this defence did not have a significant effect on the utility of the trained model. The min-max defence (i.e. the adversarial regularisation) resulted in strong protection against Type-II attack as it reduced its accuracy to 51.7%. This defence, however, was not as effective against a global loss-based attack where the accuracy reached 60.6%. We found knowledge distillation to be an effective defence as no attacks could reach accuracy over 53%. Training a victim model with DP-SGD resulted in the strongest protection against all attacks when compared to other defence methods, but at a cost of severe utility degradation as shown in Figure 5.

Table 6 summarises the effects the argmax defence, crop training, mix-up and adversarial regularisation have when applied to binary segmentation models trained on the Kvasir-SEG dataset. Here we employed Type-I, Type-II and global loss-based attacks. The best performing defences were crop-based training and adversarial regularisation, for which the attack accuracy did not exceed 53.3% and 55.3%, respectively, regardless of the attack employed.

## 6.2  Membership Inference Attacks on Multi-Class Segmentation Models

We investigate the performance of Type-I, Type-II and global loss-based membership inference attacks on multi-class segmentation models trained on a subset of the Cityscapes dataset, and compare the effects of the argmax defence, and crop-based and mix-up training regularisation. The results are summarised in Table 7. The Type-I and Type-II attack achieved similar performances regardless of whether the victim model was defended or otherwise. The highest accuracy of 85.1% was obtained by both Type-I and Type-II attacks on undefended victim models, however the Type-I attack resulted in a slightly more favourable F1-score. The argmax defence reduced success of the attack marginally for all three evaluated attack implementations. By applying a mix-up defence during training of the victim model, the attack accuracy was decreased by 7.7%,

Table 5: Comparison of Type-II and global loss-based MI attacks (in %) given different defence mechanisms used by the victim during training. The segmentation models share the same architecture and were trained on the Liver subset of MSDD. Highlighted values indicate the worst MIA performance given a chosen defence, i.e. the highest level of protection.

| Attack | Defence | Accuracy | F1-score |
|---|---|---|---|
| Type-II | - | 85.3 | 86.3 |
| | Argmax | 67.0 | 65.8 |
| | Crop training | 55.3 | 58.8 |
| | Mix-up | 63.6 | 45.8 |
| | Min-max | 51.7 | 56.1 |
| | DP ($\varepsilon = 8.5, \delta = 2e-3$) | 50.2 | 40.3 |
| | KD | 52.8 | 50.8 |
| Global loss-based | - | 80.6 | 76.2 |
| | Argmax | 75.3 | 67.4 |
| | Crop training | 50.3 | 1.2 |
| | Mix-up | 58.5 | 29.5 |
| | Min-max | 60.6 | 35.0 |
| | DP ($\varepsilon = 8.5, \delta = 2e-3$) | 49.9 | 0.0 |
| | KD | 50.2 | 0.8 |

Table 6: Comparison of Type-I, Type-II and global loss-based MI attacks (in %) given different defence mechanisms used by the victim during training. The segmentation models share the same architecture and were trained on the Kvasir-SEG dataset. Highlighted values indicate the worst MIA performance given a chosen defence, i.e. the highest level of protection.

| Attack | Defence | Accuracy | F1-score |
|---|---|---|---|
| Type-I | - | 75.8 | 75.2 |
| | Argmax | 59.8 | 56.6 |
| | Crop training | 53.3 | 13.0 |
| | Mix-up | 69.0 | 70.5 |
| | Min-max | 48.6 | 0.7 |
| Type-II | - | 93.3 | 93.2 |
| | Argmax | 89.5 | 89.4 |
| | Crop training | 51.8 | 23.7 |
| | Mix-up | 58.0 | 30.0 |
| | Min-max | 54.5 | 64.3 |
| Global loss | - | 80.5 | 80.3 |
| | Argmax | 72.3 | 61.6 |
| | Crop training | 50.5 | 4.8 |
| | Mix-up | 71.5 | 62.3 |
| | Min-max | 55.3 | 19.0 |

10.2% and 9.3% for Type-I, Type-II and global loss-based attacks, respectively. The best performing defence was crop-based training, which caused the most severe attack accuracy reduction of 35.2% for Type-I attack.

Table 7: Comparison of Type-I, Type-II and global loss-based MI attacks (in %) given different defence mechanisms used by the victim during training. The segmentation models share the same architecture and were trained on the Cityscapes dataset. Highlighted values indicate the worst MIA performance given a chosen defence, i.e. the highest level of protection.

| Attack | Defence | Accuracy | F1-score |
|--------|---------|----------|----------|
| Type-I | - | 85.1 | 84.4 |
| | Argmax | 84.3 | 84.2 |
| | Crop training | 49.9 | 65.9 |
| | Mix-up | 77.4 | 80.3 |
| Type-II | - | 85.1 | 84.1 |
| | Argmax | 84.6 | 86.2 |
| | Crop training | 54.5 | 64.7 |
| | Mix-up | 74.9 | 70.1 |
| Global loss | - | 79.8 | 77.6 |
| | Argmax | 79.3 | 77.2 |
| | Crop training | 57.0 | 37.4 |
| | Mix-up | 70.5 | 60.5 |

## 6.3 Membership Inference Attacks and Backdoor Attacks

### 6.3.1 Varying Poisoning Rates

We evaluate the attack under the threat model where an adversary has an access to the training data loader of the victim model with the aim of embedding a backdoor, and then attacking the trained model to infer the membership status of the victim data records. The performance overview for both attacks is summarised in Tables 8 and 9 for Type-II and global loss-based MIAs, respectively.

To successful embed a backdoor into the victim model, 2% of the training dataset needed to be poisoned for the attack setting that employed a line trigger. However, in the case of the square trigger, the poisoning rate needed to be at least 4%.

The accuracy of Type-II membership inference attacks peaked at 79.2% for the line trigger and at 82.1% for the square trigger in the setting where the poisoning probability was set to 5%. The performance of Type-II attacks gradually decreased when reducing the poisoning probability. However, the attack accuracy increased by 15.4% and 4.1% for the line and square triggers, respectively, once the probability reached the point at which the backdoor attack was no longer successful. We additionally note that a failure to embed a backdoor results in poisoned samples being treated as benign by the target model.

Global loss-based attack achieved the highest accuracy of 77.3% in a setting where the backdoor attack had the lowest impact, i.e. with the poisoning probability set to 1% while using a square trigger. Under a threat model where the data samples poisoned using the line trigger the top accuracy of the global loss-based attack was 75.3% for a poisoning probability of 5%. Similarly to Type-II attack, the gradual accuracy decrease caused by lowering the poisoning probability could be observed with the line trigger as well. However, the substantial performance increase caused by the backdoor attack failure was not present in such setting. For the square trigger, we did not observe any pattern of performance deviations.

### 6.3.2 Trigger Value Dependence

Table 10 shows the changes in the backdoor and membership inference attack performances when varying the trigger contrast. Embedding a backdoor was possible for trigger values larger or equal to 100 and 150 for the line and the square triggers, respectively. For experiments where the backdoor attack was successful, the highest attack accuracy of 76.2% was obtained for a square trigger with its values set to 200.

Table 8: Performance comparison of a Type-II attack (in %) on an infected model, and poisoning success given two different triggers under various poisoning rates.

| Poisoning prob. | Backdoor | Accuracy | F1-score |
|---|---|---|---|
| $1 \times 256$ trigger | | | |
| 0.05 | Yes | 79.2 | 75.5 |
| 0.04 | Yes | 75.0 | 79.5 |
| 0.03 | Yes | 64.7 | 62.7 |
| 0.02 | Yes | 60.7 | 37.5 |
| 0.01 | No | 76.1 | 80.5 |
| $3 \times 3$ trigger | | | |
| 0.05 | Yes | 82.1 | 79.1 |
| 0.04 | Yes | 73.9 | 78.6 |
| 0.03 | No | 78.0 | 75.2 |
| 0.02 | No | 71.9 | 78.1 |
| 0.01 | No | 66.6 | 74.7 |

Table 9: Performance comparison of a global loss-based attack (in %) on an infected model, and poisoning success given two different triggers under various poisoning rates.

| Poisoning prob. | Backdoor | Accuracy | F1-score |
|---|---|---|---|
| $1 \times 256$ trigger | | | |
| 0.05 | Yes | 75.3 | 67.2 |
| 0.04 | Yes | 70.5 | 58.6 |
| 0.03 | Yes | 70.4 | 58.3 |
| 0.02 | Yes | 65.7 | 48.1 |
| 0.01 | No | 64.7 | 45.8 |
| $3 \times 3$ trigger | | | |
| 0.05 | Yes | 72.0 | 61.3 |
| 0.04 | Yes | 74.4 | 65.8 |
| 0.03 | No | 71.6 | 60.3 |
| 0.02 | No | 72.3 | 61.7 |
| 0.01 | No | 77.4 | 71.0 |

# 7 Discussion

Our experimental evaluation demonstrates a high vulnerability of semantic segmentation architectures to membership inference and backdoor attacks. The ability to successfully mitigate MIAs often results in unfavourable privacy-utility trade-offs or substantially increased computational costs. We conclude our work by discussing the effectiveness of the proposed defences as well as the applicability of individual threat models to generic semantic segmentation settings.

## 7.1 Membership Inference Attacks and Defences

### 7.1.1 Attack Success

Our experimental results indicate that membership inference attack on segmentation models pose a significant threat to a large number of learning contexts. We determine that the accuracy of a MIA against an undefended model can exceed 85% in certain learning settings. For a binary segmentation setting, Type-I attacks typically yield the worst performance given very limited information in possession of the adversary.

Table 10: Comparison of backdoor attacks given varying colour of a trigger, and the associated effects on a global loss-based MI attack (in %). For all runs, the probability of poisoning each data sample during training was set to 0.05.

| Trigger value | Backdoor | Accuracy | F1-score |
|:---:|:---:|:---:|:---:|
| $1 \times 256$ trigger | | | |
| 255 | Yes | 75.3 | 67.2 |
| 200 | Yes | 73.5 | 64.1 |
| 150 | Yes | 71.0 | 59.5 |
| 100 | Yes | 72.0 | 61.4 |
| 50 | No | 72.8 | 62.6 |
| 25 | No | 70.7 | 58.6 |
| $3 \times 3$ trigger | | | |
| 255 | Yes | 72.0 | 61.3 |
| 200 | Yes | 76.2 | 69.1 |
| 150 | Yes | 69.4 | 56.0 |
| 100 | No | 73.7 | 64.4 |
| 50 | No | 78.1 | 72.2 |

Table 11: Overview of defences for binary segmentation models compared by the level of offered protection, utility effects, additional computation cost, and additional model training requirements.

| Defence | Protection | Utility decrease | Cost | Additional model |
|:---|:---:|:---:|:---:|:---:|
| Argmax | Low | Low | Low | No |
| Crop training | High | High | Low | No |
| Mix-up | Medium | Low | Low | No |
| Min-max | High | Low | High | Yes |
| DP | High | High | High | No |
| KD | High | Low | High | Yes |

By combining the predicted segmentation masks with the ground-truth masks, the adversary is able to significantly improve the the accuracy of the attack, particularly in settings where the shadow model shares the architecture with the victim model. This, however, is not the case for multi-class segmentation models where the observed difference between the performances of Type-I and Type-II attacks was minimal. Multi-class segmentation models frequently output a mask with prediction probability distributions over the set of assumed classes for each pixel. This accumulated information provides the adversary with more knowledge about the internal behaviour of the victim model, and can, therefore, make it more susceptible to membership inference attacks.

The global loss-based attack outperforms the Type-II attack in a majority of experiments with binary segmentation models where the architectures of the victim and shadow models were not identical, showing its robustness and efficiency, while also being the simplest and least computationally expensive attack to execute. As the membership inference attack can be thought of as an anomaly detection, its high performance at low false-positive rates indicates its significance and threat.

### 7.1.2 Defences and Their Effectiveness

The success of membership inference attacks can be mitigated with the use of appropriate defence mechanisms. Reducing the amount of information included in predictions of a segmentation model by outputting only the prediction labels provides very limited protection in a binary setting and almost no protection for a multi-class segmentation setting. Training the target model on random crops taken out of the training

data is effective and offers strong protection against Type-I, Type-II and global loss-based attacks. However, the utility of such a model is reduced which results in an unfavourable privacy-utility trade-off. Mix-up regularisation does not provide similar level privacy protection when compared to the crop-based training. However, if the model owner(s) need to prioritise model utility, this defence results in a more favourable protection-utility trade-off when compared against other data augmentation-based defences.

Defence methods that require training of an additional model (e.g. min-max or knowledge distillation) offer the best protection without an associated utility penalty, making them exceptionally suitable for contexts that possess the required data and compute to train the additional models. The disadvantage of adversarial regularisation is its vulnerability to global loss-based attacks. Finally, the highest level of privacy protection is offered by DP-SGD, resulting in the largest attack accuracy reduction. This is not surprising, as DP-SGD offers a formal guarantee which is in-line with the membership inference interpretation of differential privacy. However, it comes at a large utility penalty compared to the other mechanisms, making this method problematic for certain settings where such trade-off would be unacceptable.

### 7.1.3 The Applicability of Global Loss-Based Attack

The global loss-based attack is, in general, robust to the mismatch of the victim and shadow model model architectures and poses a significant threat in cases where an adversary has access to auxiliary information such as the data distribution, target model architecture and the associated hyperparameters. However, the attack is sensitive to the differences in utility between the victim and shadow models. If the victim model relies on strong regularisation during training its overall utility and the loss on testing data samples will be different compared to a shadow model that does not employ any regularisation during the training process. As the decision threshold for membership inference is obtained through the shadow model, the arising performance gap will reduce the performance of the attack.

### 7.1.4 The Threat of Overfitting

One of the fundamental reasons for the success of membership inference attacks is overfitting of the victim model (Shokri et al., 2017; Salem et al., 2018; Yeom et al., 2018; Chen et al., 2020; Leino & Fredrikson, 2020) (which is supported by the results of our work). We have shown that performance of MIAs on segmentation models increases with the size of a generalisation gap. This suggests that in cases where the amount of training data is limited or the model capacity is not sufficient for the task, defences that attempt to reduce the generalisation gap can be highly beneficial. Additionally, our experiments show that having sufficient quantity of data reduces the risks associated with MIAs as the possibility of "imprinting" the information about individual data points is severely limited. Therefore, selecting a model architecture with a reasonable learning capacity and appropriate architecture and associated hyperparameters can serve as a defence mechanism on itself (as discussed by Usynin et al. (2022) for instance).

### 7.1.5 Privacy-Utility Trade-off and Computational Cost

In our evaluation we found the application of adversarial regularisation during training and the use of knowledge distillation to result in the best privacy-utility trade-off. However, both of these defences impose additional computational requirements on the data owner as they require additional (trained) model, which may not be achievable in certain settings. Other defences that offered high levels of protection were the crop-based training and the utilisation of DP-SGD. While they do not require the data owner to train a second auxiliary model, they, nonetheless, result in utility degradation for the target model.

## 7.2 Membership Inference and Backdoor Attacks

### 7.2.1 The Threat of Backdoor Attacks

As we discovered during our experimental runs, backdoor attacks can pose a significant threat to collaboratively trained segmentation models. Furthermore, in this work we show that a trigger consisting of only nine pixels can be enough to poison the victim model and alter the results of liver segmentation (to the point where no segmentation map is generated). These results, when put into a practical perspective, can showcase

how vulnerable models deployed in critical settings (such as medical image analysis) can be. Generally, the success of a backdoor attack increases with a larger trigger, which itself increases the probability that such sample is detected. However, with large triggers the contrast between the background and the trigger can be reduced to a higher extent (as these tend to blend in much better) when compared to smaller triggers, making the selection of appropriate triggers a really attractive avenue for future work in this domain.

### 7.2.2 Membership Inference Attacks on Poisoned Models

In general, we note that the addition of backdoor attacks reduced the performance of MIAs (membership inference accuracy was highest when the adversary targeted a benign model). Membership inference attacks on poisoned models had higher accuracy in settings where the amount of poisoning information (e.g. the number of poisoned data samples or the trigger visibility) was also high, i.e. the distinction between the main-task and the backdoor utility was clearly defined. This allowed the poisoned model to maintain the same level of the main-task utility as a benign model during training, thus successfully concealing a backdoor. By reducing the number of poisoned samples or the perceptibility of the trigger, the inclusion of backdoors becomes more challenging as there was simply less data to guide the backdoor embedding process.

We observe that in cases where the poisoned data sample was not recognised (i.e. the predicted mask included the segmented object) during training, the loss was significantly higher when compared to benign image-mask pairs. For attack settings where the poisoning rate was low (and the task of embedding a backdoor was, therefore, more challenging) such loss deviations resulted in convergence problems and an overall utility decrease. This explains a higher variance of MIA accuracy on models trained with lower poisoning rates as the convergence was typically less stable during training and the utility of the trained model could be negatively affected. The Type-II attack was generally more sensitive to this effect as the difference between the highest and lowest accuracies (regardless of the trigger size) was 21.4%, whereas for the global loss-based attack it was only 12.7%.

### 7.3 Limitations & Future Work

In this paper, we primarily use the accuracy and the F1-score to evaluate the success of MIAs. While these metrics are commonly used to measure performance of binary classifiers, they do not necessarily quantify the "overall" privacy risks of membership inference. Typically, MIA performs much better on the out-of-distribution samples, as they can have a more profound contribution during target model training (Long et al., 2018). For the adversary, the ability to successfully detect member samples at low false-positive rates can be more valuable than the overall accuracy of the attack model, as it allows to effectively measure the privacy leakage in an average-case guessing setting. As there are many potential "success metrics" for MIAs, we acknowledge that there may be other means of measuring the adversarial performance, but we leave such additional evaluation as future work.

In this work, we limit ourselves to smaller datasets (or subsets thereof). By doing so, we are able to amplify the effects of overfitting and to effectively reason over the performance differences of various defence mechanisms. We acknowledge that while smaller datasets can in many cases lead to better attack results, we expect the overall trends to remain similar. While some prior work in the domain of image classification concludes that the use of large-scale datasets can serve as a "mitigation mechanism" on itself (Usynin et al., 2022), there is no well-defined threshold for the size of such dataset. As a result, we leave a detailed investigation of the relationship between the dataset size and the relative reduction in attacker's performance as a further extension of our work.

We leave the investigation of significantly more computationally expensive defences for multi-class segmentation models as future work. Additionally, the applied knowledge distillation defence removes the class prediction distributions from the targets used to train the protected model. While our protected model was still able to obtain high utility, this setting relaxed some of the assumptions of the original knowledge distillation model (Hinton et al., 2015), where the prediction distribution is key to the success of knowledge transfer for classification settings.

While we investigated a number of scenarios where the backdoor attacks were successful, certain metrics (e.g. the performance of benign data or backdoor detection accuracy) used in prior works on backdoors were not considered as we deemed these to be outside of the scope of this work. We foresee that our work can serve as a strong foundation for a follow-up investigation of the interoperability between the backdoor attacks and the MIAs in this domain (similar to Tramèr et al. (2022)).

Finally, in this work we rely on a single shadow model when performing MIAs (which was previously shown to be sufficient for many attack contexts (Salem et al., 2018; Tonni et al., 2020)). Prior work in the area (Shokri et al., 2017; Long et al., 2017; Carlini et al., 2022) suggests that using multiple shadow models can lead to better adversarial performance, but at a cost of additional computational requirements. We identify a potential performance improvement that can be gained by the adversary with access to additional computational resources and leave an in-depth investigation of this amplification to future work.

## 8  Conclusion

In this work, we evaluated a number of membership inference and backdoor attacks against semantic segmentation models on publicly available datasets. We present results under different adversarial threat models and show that most semantic segmentation settings can be vulnerable to both the in-the-network and the out-of-the-network adversaries. We note that attack settings where the adversary has access to the ground-truth segmentation masks are particularly vulnerable to MIAs, resulting in much higher adversarial performance. Additionally, we demonstrate that there exists a number of mitigation strategies that the practitioners can employ, highlighting that the data owners often have to accept unfavourable privacy-utility trade-offs or additional computation costs in order to obtain the means of protection against such attacks. We identified that knowledge distillation and adversarial regularisation, can both provide the data owners with meaningful privacy protection, while resulting in a more favourable privacy-utility trade-off when compared to other defence strategies. Additionally, we discovered that certain semantic segmentation settings can be extremely vulnerable to backdoor attacks, further highlighting the importance of careful threat modelling when designing collaborative ML learning settings. We hope that our findings can serve as a strong foundation for the future work on the intersection of practical deep learning and adversarial interference.

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
