# OpenReview forum: "Membership Inference Attacks Against Semantic Segmentation Models"
_TMLR — Rejected by TMLR_

### Review · Reviewer_M57h · 2023-03-31

**Summary Of Contributions:**

This paper presents an empirical study on membership inference attacks targeting semantic image segmentation models. Specifically, the study uses three existing attacks and five defense methods to evaluate their performance in the context of semantic segmentation. Additionally, this paper investigates the effectiveness of membership inference attacks with the assistance of backdoor attacks. The experiments are conducted on three datasets and three model architectures. The empirical results demonstrate that the performance of membership attacks is influenced by the structure of the shadow model. Furthermore, the defense methods need to consider the trade-off between protection and utility.

**Audience:**

Yes

**Claims And Evidence:**

No

**Requested Changes:**

Please see the above weaknesses section.

**Strengths And Weaknesses:**

Strengths

+ The paper addresses an important issue by investigating the susceptibility of semantic image segmentation models to membership inference attacks, especially in the medical context.
+ The attacks and defenses studied in the paper are representative and the experiments are reasonable.

Weaknesses

- The research problems addressed in the paper lack novelty. The experimental results simply confirm that semantic segmentation models are vulnerable to membership inference attacks, which has already been established in previous research. As a result, the paper does not provide any new insights or information for readers.

- The paper repeatedly emphasizes that membership inference attacks are not commonly conducted on semantic segmentation tasks, but fails to compare the difficulty of such attacks on segmentation versus classification tasks. Are the membership inference attacks easier or harder on semantic segmentation tasks compared to classification tasks? What about defenses? Are there any differences? And why?

- Section 6 presents experimental results without offering a thorough analysis of the findings. For instance, Type-II attacks were found to be more effective than Type-I attacks in Section 6.1. This paper plainly reports the results. It does not provide an explanation or a discussion on why this observation exists. What are the root causes leading to this? Does this phenomenon also happen in classification tasks?

- The presentation is not well organized. For example, there are three tables on page 8. The contents are very similar, just the attack results for different attacks. However, they take up the whole page and make the comparison across attacks hard. Figure 2 is just a duplicate of Tables 1-3. Why not just use Figure 2, which is easier to see the difference across attacks and model architectures.

---

> ### Author Response · Authors · 2023-04-21
> **Response #1 to Reviewer M57h**
>
> Q1: The research problems addressed in the paper lack novelty. The experimental results simply confirm that semantic segmentation models are vulnerable to membership inference attacks, which has already been established in previous research. As a result, the paper does not provide any new insights or information for readers.
>
> A1: We thank the reviewer for this remark, but we would like to note that, per TMLR editorial policy, novelty is not a requirement for acceptance in TMLR. While we do agree that MIAs in semantic segmentations have been previously discussed, our study provides an extended evaluation of many different attacks, defences and learning settings under MIAs, and thus providing the community with a novel contextualisation of possible vulnerabilities of this specific training context. We discuss the specific challenges of MIA in a segmentation setting in our response below and have amended the manuscript to reflect them.
>
> Q2: The paper repeatedly emphasizes that membership inference attacks are not commonly conducted on semantic segmentation tasks, but fails to compare the difficulty of such attacks on segmentation versus classification tasks. Are the membership inference attacks easier or harder on semantic segmentation tasks compared to classification tasks? What about defenses? Are there any differences? And why?
>
> A2: We thank the reviewer for this comment. We agree that some of the messages of our work could have been conveyed more directly, and will amend Sections 1, 2.1 and 3.1. Particularly, we expand our discussion on the differences between attacks in classification and segmentation settings. We discuss these in detail below. In brief:
>   - Attacks on classification often make use the of the predictive vector of the size N (where N is the number of classes)
>   - However, attacks on segmentation settings have access to a much more informative segmentation mask of the size input image. This allows the adversary to make more informed decisions regarding the membership of the data point.
>
> We hope that these changes would allow the reader to better understand the main contributions of this work.
>
> Q3: Section 6 presents experimental results without offering a thorough analysis of the findings. For instance, Type-II attacks were found to be more effective than Type-I attacks in Section 6.1. This paper plainly reports the results. It does not provide an explanation or a discussion on why this observation exists. What are the root causes leading to this? Does this phenomenon also happen in classification tasks?
>
> A3: We thank the reviewer for their input. In brief, the Type-I attack relies solely on the victim model's output to generate a prediction, while the Type-II attack concatenates the ground-truth mask with the prediction, providing the attacker with additional information that results in superior attack performance. One crucial aspect of the Type-II attack is that it can compare the victim model's output with the ground truth, which allows for easier discernment of the model's precision and confidence. This feature is particularly important because, as shown by prior research, segmentation models tend to overfit (Li et al.) and are therefore overconfident when evaluated on training data. Incorporating the ground truth into the attack process enables the attacker to evaluate how well the victim model generalises to unseen data, which can result in more accurate MIAs. These assumptions are supported by the evaluations which demonstrate a superior performance of the Type-II attack when compared to the Type-I attack.
> Additionally, the importance of the Type-II attack (specifically in a semantic segmentation context) arises from the multidimensionality of the output. In contrast to classification tasks, segmentation model outputs and targets contain spatial information that adds an extra layer that can be exploited by the attacker to infer more about the behaviour of the victim model. This multidimensionality of the output in segmentation models is in stark contrast to classification models, which output a flat logit/softmax vector that lacks any spatial information. This makes classification models unsuitable for the Type-II attack, as the attacker cannot easily distinguish between the model's performance on different regions of the input.
> To better highlight these differences, we will amend Sections 1, 3.1, 5.2 and 7.1 and explain the attack in more detail as discussed above, as well as discuss the reasons behind the superior performance of the Type-II-attack.
>
> Q4: The presentation is not well organized. For example, there are three tables on page 8. The contents are very similar, just the attack results for different attacks. However, they take up the whole page and make the comparison across attacks hard.
>
> A4: We thank the reviewer for this suggestion and have re-organized and condensed the Figures and Table in line with their suggestions.

---

> ### Author Response · Authors · 2023-04-21
> **Response #2 to Reviewer M57h**
>
> Li, Zeju, Konstantinos Kamnitsas, and Ben Glocker. "Overfitting of neural nets under class imbalance: Analysis and improvements for segmentation." Medical Image Computing and Computer Assisted Intervention–MICCAI 2019: 22nd International Conference, Shenzhen, China, October 13–17, 2019, Proceedings, Part III 22. Springer International Publishing, 2019.

---

### Review · Reviewer_TDtZ · 2023-04-09

**Summary Of Contributions:**

This paper studies the problem of membership inference attacks (MIAs) against semantic segmentation models for images. This work adapted some existing methods originally proposed for classification tasks into the segmentation ones and revealed that segmentation models are also highly vulnerable to MIAs, and then discussed some mitigation schemes and also the interaction between data poisoning and MIAs.

**Audience:**

Yes

**Broader Impact Concerns:**

This paper unveils the threats of MIA on segmentation models with in-depth analysis and also proposes countermeasures. Therefore, no significant ethical concern is present in this paper.

**Claims And Evidence:**

Yes

**Requested Changes:**

I have the following requirements:
1. If the authors disagree with my points made in weakness 1 and 2, I hope more clarifications in terms of the significance and novelty of the paper can be added.
2. Add comparison to the baseline of He et al., (2020) so as to better highlight the contribution of this paper if that baseline underperforms the proposed method in this paper (partially addressing my point in weakness 2).
3. A high level idea on why backdoor attacks can help boost MIA in general should be provided as it is still unclear to me after reading the paper. Backdoor attacks inject some backdoor triggers Into the training set so that any input with the added trigger (regardless if it belongs to the training set or not) will be misclassified into a particular wrong label. I am not sure how this property will increase the gap between training and test samples at the high level.
4. Considering the poisoning adversary is a stronger threat model and hence, it is expected to perform better than the traditional threat model in MIAs that the training data are not manipulated. I think this is also a main point that distinguishes this work from He et al., (2020). Therefore, I would expect the authors to illustrate this point, instead of showing that, in some settings, backdoor attacks even hurt MIAs, as I cannot make a good sense of it. If it is because of issues inherent to the backdoor attacks that eventually hurts MIA, then the attacker can simply switch to other poisoning attacks to boost the performance. A doable solution is either to adapt the poisoning attack proposed in Tramèr et al., (2022) or improve the inference performance when using the backdoor attacks .

**Strengths And Weaknesses:**

Strengths:
1. Semantic segmentation is an important application, yet very few works are focused on the privacy risks in these systems.
2. The paper adapted different types of attacks and also defenses, and also the impact of data poisoning attacks on MIAs.

Weakness:
1. The results are more or less expected that semantic segmentation can be vulnerable to membership inference attacks.
2. The proposed methods are based on somewhat straightforward adaptation from the existing methods in classification tasks.
3. The performance of the original method from He et al., (2020) is not used as a comparison baseline. This might highlight the contribution of this work better.
4. The backdoor attacks part is confusing to me, as it considers a stronger threat model but the attack success is not always guaranteed to succeed. Also, it its still unclear why poising with backdoor attacks is expected to improve MIA in general.

---

> ### Author Response · Authors · 2023-04-21
> **Response #1 to Reviewer TDtZ**
>
> Q1: The results are more or less expected that semantic segmentation can be vulnerable to membership inference attacks. The proposed methods are based on somewhat straightforward adaptation from the existing methods in classification tasks. If the authors disagree with my points made in weakness 1 and 2, I hope more clarifications in terms of the significance and novelty of the paper can be added.
>
> A1: We thank the reviewer for this comment. We agree that some of the messages of our work could have been conveyed better and will amend Sections 1, 2.1 and 3.1. Concretely, we expand our discussion on the differences between attacks in classification and segmentation settings. These changes will allow the readers to better understand the main contributions of this work. We explain the concrete changes in more detail in our responses below.
>
> We additionally note that novelty is not a requirement for acceptance according to the TMLR editorial policy, and while we do agree that MIAs in semantic segmentations have been previously discussed, our study provides an extended evaluation of many different attacks, defences and learning settings under MIAs, providing the community with a contextualisation of of possible vulnerabilities of such training contexts.
>
> Lastly, we note that our work is not an adaptation of prior works such as He et al., but rather a more comprehensive study of stronger adversarial models of attacks and defences in the context of semantic segmentation. We address the core differences between the two works in our response below. In brief: we evaluate a number of model- and data-agnostic attacks and defences (contrary to the somewhat dataset-specific approaches in He et al.), allowing the research community to easily contextualise and integrate these into their learning settings, independent of architecture and/or dataset. We additionally explore more sophisticated attacks (e.g. the Type-II attack) and more permissive threat models (dishonest backdoor adversaries) to put our findings into a broader context.
>
> Q2: Add comparison to the baseline of He et al., (2020) so as to better highlight the contribution of this paper if that baseline underperforms the proposed method in this paper (partially addressing my point in weakness 2).
>
> A2: We thank the reviewer for this comment and agree that our work can be more explicit regarding the differences between our contributions and those of He et al. The work by He et al., is mostly concerned with attacks which rely on information contained in individual patches of the images and attempt to identify what constitutes an informative mask-image pairing for the adversary and which defences can be used to mitigate this effect. Moreover, it is not completely clear how well the results on patches by He et al. generalise beyond the datasets they present in their work. In our work we consider a larger number of attacks (including dataset-agnostic ones); more defences than He et al. and other prior works, most of which can be applied irrespective of the model architecture or dataset used in a learning task as well as an additional threat model with an active dishonest adversary. Overall, our work is an extension of the work by He et al., applicable to a larger number of training settings (particularly collaborative ones) and serves as a useful reference and an insight into which defences can be applicable against more sophisticated attacks (such as the Type-II attack, for instance).
> We amended Sections 2.1 and 3.1 to reflect these differences and allow the reader to differentiate between the contributions of these two works.
>
> He, Yang, et al. "Segmentations-leak: Membership inference attacks and defenses in semantic image segmentation." Computer Vision–ECCV 2020: 16th European Conference, Glasgow, UK, August 23–28, 2020, Proceedings, Part XXIII 16. Springer International Publishing, 2020.

---

> ### Author Response · Authors · 2023-04-21
> **Response #2 to Reviewer TDtZ**
>
> Q3: A high level idea on why backdoor attacks can help boost MIA in general should be provided as it is still unclear to me after reading the paper. Backdoor attacks inject some backdoor triggers Into the training set so that any input with the added trigger (regardless if it belongs to the training set or not) will be misclassified into a particular wrong label. I am not sure how this property will increase the gap between training and test samples at the high level.
>
> A3: We thank the reviewer for this remark. As seen in the work by Tramer et al., injection of adversarially modified data into the training dataset can lead to more favourable adversarial performance on multiple types of privacy-oriented attacks (such as MIA or attribute inference for instance). The use of backdoors in our work forces the model to associate a particular part of the image (i.e. the triggers) with the adversarially-controlled behaviour. In general, this behaviour can either reduce the utility of the model (overall or on a particular subset of data), but it can also be used to achieve other adversarial outcomes (such as subject reidentification in the work of Bagdasaryan et al.), including the improved privacy-oriented attack performance. Some ways of achieving this would be to force the model to memorise particular training samples by:
>   - reducing the utility of the model on the samples of interest in a collaborative setting in order to force the other party to use data which is of higher quality and may contain more revealing features, leading to their memorisation (such as the method used in the work by Zhang et al.) or
>  - through the ‘privacy onion effect’ described by Carlini et al. (where reduction in memorisation of particularly sensitive samples results in higher memorisation of another subset of samples instead).
> We agree that some of these points are not comprehensively explained and may be unobvious to the general audience. We thus amended Sections 1 and 4, where we now include a detailed explanation of adversarial motivations, favourable/vulnerable learning settings and potential threats to participants of collaborative learning settings.
>
>
> Bagdasaryan, Eugene, and Vitaly Shmatikov. "Blind backdoors in deep learning models." Usenix Security. 2021.
>
> Carlini, Nicholas, et al. "The privacy onion effect: Memorization is relative." Advances in Neural Information Processing Systems 35 (2022): 13263-13276.
>
> Zhang, Yuheng, et al. "The secret revealer: Generative model-inversion attacks against deep neural networks." Proceedings of the IEEE/CVF conference on computer vision and pattern recognition. 2020.
>
> Tramèr, Florian, et al. "Truth serum: Poisoning machine learning models to reveal their secrets." Proceedings of the 2022 ACM SIGSAC Conference on Computer and Communications Security. 2022.

---

> ### Author Response · Authors · 2023-04-21
> **Response #3 to Reviewer TDtZ**
>
> Q4: Considering the poisoning adversary is a stronger threat model and hence, it is expected to perform better than the traditional threat model in MIAs that the training data are not manipulated. I think this is also a main point that distinguishes this work from He et al., (2020). Therefore, I would expect the authors to illustrate this point, instead of showing that, in some settings, backdoor attacks even hurt MIAs, as I cannot make a good sense of it. If it is because of issues inherent to the backdoor attacks that eventually hurts MIA, then the attacker can simply switch to other poisoning attacks to boost the performance. A doable solution is either to adapt the poisoning attack proposed in Tramèr et al., (2022) or improve the inference performance when using the backdoor attacks.
>
> A4: We thank the reviewer for this comment. In general, we agree that an additional utility-oriented attack such as a backdoor insertion ‘should’ lead to a better MIA performance, as otherwise this could defeat the purpose of investing additional adversarial resources into conducting auxiliary attacks. However, the finding that such attacks in reality often end up deteriorating the inference performance instead is not specific to our work. For example, Tramer et al. also reached the conclusion that such attacks are often very challenging to execute in reality in an untargeted, model and data agnostic manner. In the work of Tramer et al., this is partially alleviated by performing many cross-validations across 128 shadow models, resulting in untargeted settings which do indeed improve MIA performance. In our work, however, such extensive validation was impossible to carry out due to computational limitations as well as the inherent complexities of backdoors for segmentation settings.
> Concretely: as per Section 4.4 in Tramer et al., selecting a valid poisoning strategy (and the associated adversarial parameters) is a challenging task on its own in an untargeted setting. Moreover, this is exacerbated by the fact that our perturbation space is significantly larger than the one used in prior work. For instance, in Tramer et al. the space is spanned by a relatively low-dimensional logit or softmax confidence vector associated with a particular sample, whereas in our work it is much higher dimensional, e.g. a 256x256 segmentation mask.
> Finally, we also highlight that –since backdoors are data-agnostic and the pattern remains unchanged throughout the attack (unlike most white-box model poisoning methods, which rely on selecting the best image perturbation at train time)-- identifying a ‘one-fits-all’ adversarial perturbation may not always be possible for an adversary. And the opposite is true as well: some of these patterns can be used to reduce the effectiveness of privacy attacks, but it is not clear how to achieve that without a significant utility penalty for the defending side, thus leaving this an open research challenge for both the attacker and the defenders, supported by the results that we show in our work.
>
> Tramèr, Florian, et al. "Truth serum: Poisoning machine learning models to reveal their secrets." Proceedings of the 2022 ACM SIGSAC Conference on Computer and Communications Security. 2022.
>
> He, Yang, et al. "Segmentations-leak: Membership inference attacks and defenses in semantic image segmentation." Computer Vision–ECCV 2020: 16th European Conference, Glasgow, UK, August 23–28, 2020, Proceedings, Part XXIII 16. Springer International Publishing, 2020.

---

### Review · Reviewer_ZHxM · 2023-04-09

**Summary Of Contributions:**

This paper studies the vulnerability of semantic segmentation models to membership inference attacks. The paper empirically approaches this problem by conducting membership attack experiments with three different attackers over binary and multi-class settings. The paper finds that the semantic segmentation models show the same level of membership risks. The paper also evaluates the impact of backdoor poisoning on the attack success and shows that, with 2-4% of poisoning samples blended into the training data, the attack success increases from 4-15%.


Contributions:

1. The paper conducts three different membership attacks on semantic segmentation models, which extends the knowledge gained from work by He et al.
2. The paper examines the effectiveness of existing defenses against membership inference attacks in semantic segmentation settings.
3. The paper studies the impact of data poisoning (or backdoor poisoning) on the success of membership inference in semantic segmentation settings.


**Audience:**

Yes

**Broader Impact Concerns:**

No concerns about the broader impacts.

**Claims And Evidence:**

No

**Requested Changes:**

1. Clarification of the knowledge gap this study addresses
2. Clarification of new findings over the prior work by He et al.
3. Comprehensive evaluation with various semantic segmentation tasks in the real-world (to generalize the observations the paper had from a few tasks)
4. Clarification of the motivation for backdooring
5. Revision of the paper writing for coherency

**Strengths And Weaknesses:**

Strengths:

1. The paper performs some experiments to evaluate the attack's success in semantic segmentation settings.
2. The paper studies the impact of backdoor poisoning on the membership inference attack’s success.


Weaknesses:

1. It’s a bit unclear what is the knowledge gap this empirical study addresses. Most experimental results confirm the previous paper’s observations.
2. I am a bit worried that this paper lacks comprehensiveness in evaluation. In particular, this paper positions as an empirical study, so I’d like to see more semantic segmentation tasks and recent models, like Facebook’s Detectron, etc.
3. It’s also unclear what is unique about backdoor attacks: what’s the difference in impacts between poisoning attacks conducted by Tramer et al. and this paper? I am worried that the paper just swaps data poisoning for backdooring.
4. The paper is not coherent: this paper starts with the knowledge gap in attacking semantic segmentation models and then in conclusion, claims that “evaluated membership inference and backdoor attacks against…”

---

> ### Author Response · Authors · 2023-04-21
> **Response #1 to ReviewerZHxM**
>
> Q1: It’s a bit unclear what is the knowledge gap this empirical study addresses. Most experimental results confirm the previous paper’s observations.
>
> A1: We thank the reviewer for this comment. We agree that some of the messages of our work could have been conveyed better and will amend Sections 1, 2.1 and 3.1. Particularly, we expand our discussion on the differences between attacks in classification and segmentation settings. We hope that these changes would allow the reader to better understand the main contributions of this work. We explain these changes in more detail in our responses below.
>
> Q2: I am a bit worried that this paper lacks comprehensiveness in evaluation. In particular, this paper positions as an empirical study, so I’d like to see more semantic segmentation tasks and recent models, like Facebook’s Detectron, etc.
>
> A2: We thank the reviewer for these suggestions. We would like to note that Detectron primarily targets instance and panoptic segmentation tasks, whereas our work concentrates on semantic segmentation. As the reviewer suggested, we carried out a number of additional experiments (whose results are in line with our previously reported results) on additional model architectures to further strengthen our experimental evaluation. We also amended section 6, in order to show our new experimental results.
>
> MIAs, Kvasir-SEG, no defence, resnet34
> | Victim model | Shadow model | Attack      | Accuracy | F1-score |
> |--------------|--------------|-------------|----------|----------|
> | PSPNet       | PSPNet       | Type-I      | 0.9065   | 0.9096   |
> | PSPNet       | PSPNet       | Type-II     | 0.9263   | 0.9298   |
> | PSPNet       | PSPNet       | Global-loss | 0.6750   | 0.5185   |
> | PAN          | PAN          | Type-I      | 0.7605   | 0.7772   |
> | PAN          | PAN          | Type-II     | 0.7768   | 0.8069   |
> | PAN          | PAN          | Global-loss | 0.7275   | 0.6305   |
> | DeepLabV3+   | DeepLabV3+   | Type-I      | 0.7128   | 0.7199   |
> | DeepLabV3+   | DeepLabV3+   | Type-II     | 0.7690   | 0.7740   |
> | DeepLabV3+   | DeepLabV3+   | Global-loss | 0.6900   | 0.5571   |
> | FPN          | FPN          | Type-I      | 0.6352   | 0.5668   |
> | FPN          | FPN          | Type-II     | 0.6502   | 0.6410   |
> | FPN          | FPN          | Global-loss | 0.7750   | 0.8060   |
>
> MIAs, Kvasir-SEG, crop defence, FPN resnet34
> | Attack      | Accuracy | F1-score |
> |--------------|--------------|-------------|
> | Type-I      	| 0.5650 	| 0.5628 	|
> | Type-II     	| 0.4902 	| 0.0325 	|
> | Global-loss 	| 0.5425 	| 0.1644 	|
>
> MIAs, Kvasir-SEG, min-max defence, FPN resnet34
> | Attack      | Accuracy | F1-score |
> |--------------|--------------|-------------|
> | Type-I      	| 0.5650 	| 0.4387 	|
> | Type-II     	| 0.5275 	| 0.2703 	|
> | Global-loss 	| 0.5062 	| 0.0606 	|

---

> ### Author Response · Authors · 2023-04-21
> **Response #2 to ReviewerZHxM**
>
> Q3: It’s also unclear what is unique about backdoor attacks: what’s the difference in impacts between poisoning attacks conducted by Tramer et al. and this paper? I am worried that the paper just swaps data poisoning for backdooring.
>
> A3: We thank the reviewer for this remark and would like to stress that there are multiple fundamental differences between our work and the work of Tramer et at.
> In Tramer et al. the images are not modified in any way, only the labels get changed in a data-dependent manner (i.e. each label is changed to the most likely incorrect label if the image was perturbed to lie on the incorrect side of the closest decision boundary).
> Concretely: Our poisoning attack only tampers with a target’s label 𝑦, while leaving
> the example 𝑥 unchanged.
> In our work, we use a pre-defined, deterministic pattern which is applied to the image at train time, as well as a modified segmentation mask (i.e. the label in segmentation tasks).
> Our method can be particularly useful during collaborative training, because our approach is both data- and model-agnostic as the pattern is always the same regardless of the image.
> Secondly, and we argue that this is the fundamental difference between the two works: Tramer et al. is positioned as the work that employs model poisoning in order to improve the results of privacy attacks (namely MIA, attribute inference and data/canary extraction). Our work, in contrast, is primarily aiming to investigate attacks and defences for MIA in semantic segmentation. Usage of backdoors allows us to achieve two things:
>   - contextualise the potential danger of MIA on semantic segmentation models which are trained in untrusted environments and
>   - outline that while such attacks can often result in better MIA performance, a lot of care needs to be taken by the adversary in order to design an attack which is both model- and data-agnostic and allows them to obtain better performance on MIAs. The latter is in line with Tramer et al., concretely section 4.4, where they discuss the complexity of such attacks in practice. Our backdoor attack is significantly more challenging to tune than its classification counterpart, as the label space is significantly larger (256x256 mask, instead of a single logit/softmax vector).
>
> Q4: The paper is not coherent: this paper starts with the knowledge gap in attacking semantic segmentation models and then in conclusion, claims that “evaluated membership inference and backdoor attacks against…”
>
> A4: We thank the reviewer for this comment. Our work is primarily aiming to showcase the dangers of MIAs under different threat models (including, but not limited to dishonest adversaries capable of modifying the training data). We agree that the focus of the work could be conveyed more clearly, and we amended Sections 1, 2.1, 2.2 (where we compare our work to prior works in the field) in order to better highlight the main contributions of our work. As the reviewer suggests, we now highlight that the use of backdoors to improve MIAs is not the target of our work; we merely demonstrate that backdoors allow transferring our findings regarding MIAs into a low-trust collaborative environment, where the adversary can have additional dishonest capabilities. As per our response above (and the work of Tramer et al.) such untargeted backdoors are often challenging to properly configure and do not necessarily lead to adversarial benefit. In the work of Tramer et al., this is alleviated by performing many cross-validations across 128 shadow models and selecting the best performing result.
>
> Tramèr, Florian, et al. "Truth serum: Poisoning machine learning models to reveal their secrets." Proceedings of the 2022 ACM SIGSAC Conference on Computer and Communications Security. 2022.

---

> ### Author Response · Authors · 2023-04-21
> **Response #3 to ReviewerZHxM**
>
> Q5: Clarification of new findings over the prior work by He et al.
>
> A5: We thank the reviewer for this remark. While in certain ways our work is similar to the work of He et al., (e.g. the overall MIA in semantic segmentation setting, some of the defences etc.), our study is much more data, model and context agnostic. The work by He et al., is primarily concerned with attacks which rely on information contained in individual patches of the images and attempt to identify what constitutes an informative mask-image pairing for the adversary and which defences can be used to mitigate this effect. Moreover, it is not completely clear how well the results on patches by He et al. generalise beyond the datasets they present in their work.
> In our work we consider a larger number of attacks (especially data-agnostic ones); more defences than the prior work, most of which can be applied irrespective of the model  architecture or dataset used in the learning task as well as an additional threat model with an active dishonest adversary. Therefore we would argue that overall our work can be seen as an extension and generalisation of the work by He et al., applicable to a larger number of training settings (particularly collaborative ones) and datasets, and serve as a useful reference and an insight into which defences are applicable against more sophisticated attacks (such as the Type-II attack, for instance).
>
> He, Yang, et al. "Segmentations-leak: Membership inference attacks and defenses in semantic image segmentation." Computer Vision–ECCV 2020: 16th European Conference, Glasgow, UK, August 23–28, 2020, Proceedings, Part XXIII 16. Springer International Publishing, 2020.

---

### Decision · Action_Editors · 2023-06-04

**Recommendation:** Reject

**Comment:**

Reviewer M57h says that the paper is limited in insights that would advance the field and offer fresh perspectives and gives a Reject.
Reviewer ZHxM and TDtZ both gave Leaning Accepts but were hesitant in their decisions, citing lack of satisfaction with the paper in its current state.

Given my comments above for Claims and Audience, I believe this paper has potential be published at TMLR but in its current form has some major revisions it needs to address. For example, more can be added in terms of connecting the MIA objective with the backdooring threat model so that it can be of interest to a wider audience. I therefore opt for a reject at this time.

**Audience:**

Given that this paper does not have much novelty in terms of the attack methodology, and rather focuses on doing comprehensive experiments on MIA on semantic segmentation, I feel like it is not of interest to the general security and privacy community. It is, however, of interest for anyone at the intersection of semantic segmentation and privacy. This can include some folks in the medical imaging community and other communities interested in semantic segmentation where private data is of concern. Overall this may be a limited community.

From the reviews, there was also a decent amount of interest from ZHxM and TDtZ on the backdooring attack. The authors claim that by introducing some backdoor examples into the training set, they can cause the MIA attacks to be more successful. This was first shown in Tramer et al (2022). The authors did not contribute much by way of methodology here, but rather as TDtZ says, made a "very simple adoption of backdoor attacks without any attempt in incorporating the adversarial goal of successful MIAs, which makes the whole result on backdoor attacks not very exciting." I also agree that this undersells the potential connecting the backdooring attack with the MIA objective. Thus it misses an opportunity to be of greater interest for the security and privacy community.

**Claims And Evidence:**

The authors claim they address knowledge gap of the lack of understanding and research on membership inference attacks and defenses specifically in the domain of semantic image segmentation. However all reviewers are unsure what knowledge gap is addressed, noting that the MIA vulnerability results for semantic segmentation are to be expected and were already shown in previous work. The authors rebut that they're providing a comprehensive evaluation of MIA attacks and defenses for semantic segmentation, "aiming to showcase the dangers of MIAs under different threat models." Given that the classification and segmentation tasks are similar, and thus the results are to be expected, I still feel that the so-called knowledge gap that's addressed in this paper is rather small.

Reviewer M57h noted that there was no novelty in the paper and reviewer TDtZ said the methods were a straightforward adaptation of previous methods on classification onto semantic segmentation. Since the review criteria of TMLR does not require novelty, I will disregard these criticisms.

**Resubmission Of Major Revision:**

The authors may consider submitting a major revision at a later time.